# Using Unilateral Strength, Power and Reactive Strength Tests to Detect the Magnitude and Direction of Asymmetry: A Test-Retest Design

**DOI:** 10.3390/sports7030058

**Published:** 2019-03-04

**Authors:** Chris Bishop, Paul Read, Shyam Chavda, Paul Jarvis, Anthony Turner

**Affiliations:** 1Faculty of Science and Technology, London Sports Institute, Middlesex University, London NW1 4RL, UK; C.Bishop@mdx.ac.uk (C.B.); S.Chavda@mdx.ac.uk (S.C.); P.Jarvis@mdx.ac.uk (P.J.); A.N.Turner@mdx.ac.uk (A.T.); 2Athlete Health and Research Performance Center, Aspetar Orthopaedic and Sports Medicine Hospital, Doha 29222, Qatar

**Keywords:** Inter-limb differences, limb dominance, variability

## Abstract

The aims of the present study were to determine test-retest reliability for unilateral strength and power tests used to quantify asymmetry and determine the consistency of both the magnitude and direction of asymmetry between test sessions. Twenty-eight recreational trained sport athletes performed unilateral isometric squat, countermovement jump (CMJ) and drop jump (DJ) tests over two test sessions. Inter-limb asymmetry was calculated from both the best trial and as an average of three trials for each test. Test reliability was computed using the intraclass correlation coefficient (ICC), coefficient of variation (CV) and standard error of measurement (SEM). In addition, paired samples *t*-tests were used to determine systematic bias between test sessions and Kappa coefficients to report how consistently asymmetry favoured the same side. Within and between-session reliability ranged from moderate to excellent (ICC range = 0.70–0.96) and CV values ranged from 3.7–13.7% across tests. Significant differences in asymmetry between test sessions were seen for impulse during the isometric squat (*p* = 0.04; effect size = –0.60) but only when calculating from the best trial. When computing the direction of asymmetry across test sessions, levels of agreement were fair to substantial for the isometric squat (Kappa = 0.29–0.64), substantial for the CMJ (Kappa = 0.64–0.66) and fair to moderate for the DJ (Kappa = 0.36–0.56). These results show that when asymmetry is computed between test sessions, the group mean is generally devoid of systematic bias; however, the direction of asymmetry shows greater variability and is often inter-changeable. Thus, practitioners should consider both the direction and magnitude of asymmetry when monitoring inter-limb differences in healthy athlete populations.

## 1. Introduction

Inter-limb asymmetry refers to differences in the performance or function of one limb with respect to the other [1,2]. Strength and jumping-based tests are often used to quantify these differences when assessing the physical characteristics of athletes [3,4,5], largely because these are considered fundamental physical qualities to enhance athletic performance. Strength testing methods to quantify asymmetry have included the back squat [5,6], isometric squat and mid-thigh pull [7,8] or isokinetic dynamometry [9,10]. Jump tests such as countermovement jumps (CMJ) [3,4,5,11,12] and drop jumps (DJ) [13,14] are also commonly assessed to quantify asymmetry, most likely because of their similarity to sport-specific movement patterns, ease of implementation and time-efficient nature.

When asymmetry is considered, more affordable versions of force platforms are available compared to 10–15 years ago; thus, assessments of between-limb differences using the force-time curve are now a practically viable option for a wide range of athletes [15]. For example, when considering jump tests, previous research has highlighted the importance of additional metrics beyond jump height such as peak and mean force and propulsive and braking impulse [16,17], because they allow some interpretation of jump strategy rather than outcome measures alone. However, limited literature exists in this capacity with respect to asymmetry; therefore, further examination of unilateral tests which can be used to quantify inter-limb differences over more than a single test session is warranted [18,19].

Regardless of the test selected, another consideration for asymmetry is how the data are reported. Typically, testing protocols encourage three trials [20], with some studies quantifying asymmetry from the best trial [8,21] and others from the average of all trials performed [3,14]. To the authors’ knowledge, no study has directly compared asymmetry scores when calculating the percentage difference between limbs from the best score and an average of all test trials. Given previous literature has suggested the variable nature of asymmetry [11,13,21], it is plausible that these methods would result in notable differences in the magnitude of asymmetry. Thus, examining whether significant differences exist between test sessions and calculation methods (best versus average) would provide practitioners with meaningful information as to which method might be favourable for inter-limb asymmetry profiling.

Literature on this topic has also highlighted the task-specific nature of asymmetry [11,14,21,22,23]. Impellizzeri et al. [24] and Maloney [25] have discussed the concept of the ‘direction of asymmetry’ which provides an indication as to which limb may be dominant, for example, during a jump test. Recent literature has shown that the direction of asymmetry may be just as variable as the magnitude [11,14,23,26]. Bishop et al. [11] used the unilateral isometric squat, CMJ and broad jump to detect how consistently peak force (PF) and impulse favoured the same limb across tests using a Kappa coefficient. With the exception of propulsive impulse between jumps, levels of agreement ranged from slight to fair (Kappa range = −0.34 to 0.32), indicating that the direction of asymmetry varied substantially between tests. Whilst useful, the above studies reporting the direction of asymmetry have only done so for a single test session. Thus, further information regarding how consistent the direction of asymmetry is across more than a single test session is again warranted.

Cumulatively, the available evidence indicates that further research is required to examine more metrics during unilateral tasks, if the best versus average asymmetry score is more reliable for test re-test comparison and if there is consistency in the direction of asymmetry between sessions. Therefore, the aims of the present study were threefold: (1) to determine the test-retest reliability of unilateral strength and jumping-based tests that can be used to quantify asymmetries, (2) determine whether any significant differences exist for asymmetry between test sessions and, (3) determine how consistently asymmetries favour the same side between tests sessions.

## 2. Materials and Methods

### 2.1. Study Design

This study used a test-retest design enabling both within and between-session data to be quantified for three unilateral tests: the isometric squat, CMJ and DJ. Test reliability was computed three ways and inter-limb asymmetries were calculated from two methods: the best trial and as an average of all trials. Systematic bias was quantified between test sessions to determine any significant changes in test scores and asymmetry values. Finally, Kappa Coefficients were used to determine the levels of agreement for the direction of asymmetry [11], showing whether the same limb scored higher between test sessions.

### 2.2. Participants

Twenty-eight recreational soccer and rugby athletes (age = 27.29 ± 4.6 years; mass = 80.72 ± 9.26 kg; height = 1.81 ± 0.06 m) volunteered to take part in this study. Inclusion criteria required all participants to have a minimum of two year’s resistance training experience, with any participant excluded from the study if they had experienced a lower body injury at the time of testing or in the preceding three months. Participants read and were required to provide written informed consent forms to demonstrate that they were willing and able to undertake all testing protocols. Ethical approval was granted from the London Sports Institute Research and Ethics committee at Middlesex University.

### 2.3. Procedures

Participants visited the laboratory twice and performed three trials on each limb for the following tests during both visits: unilateral isometric squats, CMJ and DJ on a single force platform (PASPORT force plate, PASCO Scientific, Roseville, CA, USA) sampling at 1000 Hz. Test order was randomized so as to minimize potential fatigue impacting one specific test, although the same test order was retained during both test sessions for each participant. A familiarization session was conducted 72 h before the first data collection session, so as to reduce any potential learning effects during data collection sessions. Participants were provided with the relevant test instructions and the opportunity to practice each assessment until they reached a satisfactory level of technical competence, which was monitored throughout by an accredited strength and conditioning coach. A standardized dynamic warm up was conducted prior to each session consisting of dynamic stretches to the lower body, in addition to three practice trials at approximately 60%, 80% and 100% of perceived maximal effort for all tests. Three minutes of rest was provided after the final warm up trial before undertaking the first test and test sessions were separated by a minimum of 72 h.

#### 2.3.1. Unilateral Isometric Squat

A custom built isometric testing system (Absolute Performance, Cardiff, UK) was used for this test protocol (Figure 1a,b). Firstly, participants were instructed to step on to the centre of the force plate with their foot pointing forward. A goniometer was used to measure 140° of hip and knee flexion [8,18] for each participant, with full extension of the knee joint equalling 180°. The fulcrum of the goniometer was positioned on the lateral epicondyle of the femur. The stabilisation arm was lined up along the line of the fibula (in the direction of the lateral malleolus) and the movement arm was lined up with the femur (pointing towards the greater trochanter at the hip). The non-stance limb was required to hover next to the working limb, so as to try and keep the hips level during the isometric squat action; thus, aiding balance and stability. Once in position, participants were required to remain motionless for 2 s, without applying any upwards force (which was verified by manual detection of the force-time curve in real time). Each trial was then initiated by a “3, 2, 1, Go” countdown and participants were instructed to try and extend their knees and hips by driving up as “fast and hard as possible” [7,27] against the bar for five seconds. Recorded metrics for each trial included PF and impulse at 0.3 s, which was chosen as the specified epoch for impulse based on comparable research using the unilateral mid-thigh pull [7]. The first meaningful change in force was established when values surpassed five standard deviations (SD) of each participant’s body mass [28]. Peak force was defined as the maximum force generated during the test and impulse was defined as the net force multiplied by the time taken to produce it at 0.3 s; that is, the area under the force-time curve [7].

#### 2.3.2. Unilateral Countermovement Jump

Participants were instructed to step onto the centre of the force plate (foot pointing forward) with their designated test leg with hands placed on hips and were required to remain in the same position for the duration of the test. Due to the portable nature of the force platform, weight plates were positioned on the ground, touching each side of the force platform to ensure no movement occurred throughout testing. The jump was initiated by performing a countermovement to a self-selected depth before accelerating vertically as explosively as possible into the air. The test leg was required to remain fully extended throughout the flight phase of the jump before landing back onto the force plate as per the set up. The non-jumping leg was slightly flexed with the foot hovering at mid-shin level and no additional swinging of this leg was allowed during trials. Each trial was separated by 60 s of rest. Recorded metrics included jump height and peak propulsive force, with definitions for their quantification conducted in line with suggestions by Gathercole et al. [17] and Chavda et al. [29]. Jump height was defined as the maximum height achieved calculated from the impulse-momentum method. Peak propulsive force was defined as the maximum force output during the propulsive phase of the jump and was determined when a positive centre of mass velocity was achieved [29].

#### 2.3.3. Unilateral Drop Jump

Participants started by standing on an 18 cm box which was chosen as the height to drop from based on previous research [13,14]. With hands fixed on hips, participants were required to step off the box with their designated test leg which subsequently landed on the centre of the force plate below. Upon landing, participants were instructed to minimize ground contact time and jump as high as possible thereafter in line with previous DJ research [13,14]. Each trial was separated by a 60 s rest period and recorded metrics included jump height (calculated from the flight time method) and reactive strength index (RSI), quantified using the equation flight time/ground contact time [14].

### 2.4. Statistical Analysis

Initially all force-time data were exported to Microsoft Excel™, expressed as means and standard deviations (SD) and later transferred into SPSS (V.24, Chicago, IL, USA) for additional analyses. Within-session reliability was quantified using the coefficient of variation (CV) inclusive of 95% confidence intervals, standard error of the measurement (SEM) and a 2-way random ICC with absolute agreement inclusive of 95% confidence intervals [30]. The CV was calculated via the formula: (SD(trials 1–3)/average(trials 1–3) × 100) with values ≤ 10% suggested to be considered acceptable [16]. ICC values were interpreted in line with suggestions by Koo and Li, [31] where scores >0.9 = excellent, 0.75–0.9 = good, 0.5–0.75 = moderate and <0.5 = poor. The SEM was calculated using the formula: SD × √(1 − ICC) [31,32]. For between-session reliability, best scores were used to calculate a CV and ICC value as previously described. Inter-limb asymmetries were quantified as a percentage difference between limbs (from either best trials or an average of all trials on each side) using the formula: (100/(maximum value)*(minimum value)* − 1 + 100), as proposed by Bishop et al. [11]. When depicting inter-limb differences individually, the use of an ‘IF function’ in Microsoft Excel was added on the end of the formula: *IF (left < right, 1, −1) [11], in order to show the direction of asymmetry without altering the magnitude. To determine systematic bias, paired samples *t*-tests were conducted to quantify whether test or asymmetry scores were significantly different between sessions, with statistical significance set at *p* < 0.05. Magnitude of change was quantified between test sessions using Hedge’s *g* effect sizes: (Mean_session1_ − Mean_session2_)/SD_pooled_. These values were interpreted in line with a suggested scale by Hopkins et al. [33] where <0.20 = trivial; 0.20–0.60 = small; 0.61–1.20 = moderate; 1.21–2.0 = large; 2.01–4.0 = very large. Finally, Kappa coefficients were calculated to determine the levels of agreement for how consistently an asymmetry favoured the same side; thus, providing the ‘direction of asymmetry.’ This method was chosen because the Kappa coefficient describes the proportion of agreement between two methods after any agreement by chance has been removed [34]. Kappa values were interpreted in line with suggestions from Viera and Garrett [35], where 0.01–0.20 = slight, 0.21–0.40 = fair, 0.41–0.60 = moderate, 0.61–0.80 = substantial and 0.81–0.99 = almost perfect.

## 3. Results

Within-session reliability data are presented in Table 1. The isometric squat showed good to excellent relative reliability during both test sessions (ICC = 0.89–0.96) but also the greatest variability of all tests (CV = 4.9–13.7%), although PF showed low variability during both test sessions (CV ≤ 5.7%). The unilateral CMJ showed good to excellent reliability and acceptable variability in both test sessions (ICC = 0.81–0.93; CV ≤ 5.8%). The unilateral DJ showed good to excellent reliability and acceptable variability in both test sessions (ICC = 0.78–0.94; CV ≤ 8.1%). Between-session reliability data followed a similar trend to the within-session results. The isometric squat showed good to excellent reliability (ICC = 0.85–0.93) and the greatest variability of all tests (CV = 6.4–12.9%). The unilateral CMJ showed good reliability and acceptable variability for all metrics (ICC = 0.78–0.85; CV ≤ 6.3%). Finally, the unilateral DJ showed moderate to good reliability and slightly higher variability between sessions than the CMJ test (ICC = 0.70–0.84; CV ≤ 11.2%).

Descriptive data and inter-limb asymmetry scores are presented in Table 2. Results from the paired samples *t*-tests showed a significant difference in asymmetry was seen between sessions for impulse during the isometric squat (*p* = 0.04); however, this was only when calculating asymmetries from the best trial method. No other significant differences in asymmetry were present between sessions.

Between-session effect size data for test and asymmetry scores (quantified using both methods) are shown in Table 3. Trivial to small effect sizes were evident for test and asymmetry scores when calculated from both the best and average of all trial methods. Notably, the largest effect size was shown for impulse asymmetry (−0.60) during the isometric squat test, when calculated from the best trial method.

Levels of agreement for asymmetry scores between test sessions were calculated using the Kappa coefficient and are shown in Table 4. Results showed levels of agreement between test sessions were fair to substantial for the isometric squat test (Kappa range = 0.29–0.64), substantial for the CMJ (Kappa range = 0.64–0.66) and fair to moderate for the DJ (Kappa range = 0.36–0.56). Given the changing nature of the direction of asymmetry between test sessions, individual asymmetry data are presented in Figure 2, Figure 3 and Figure 4.

## 4. Discussion

The aims of the present study were threefold: (1) to determine the test-retest reliability of unilateral strength and jumping-based tests that can be used to quantify asymmetries, (2) determine whether any significant differences exist for asymmetry between test sessions when calculating differences from the best trial and an average of all trials and, (3) determine how consistently asymmetries favour the same side between tests sessions. Results showed moderate to excellent reliability for all tests both within and between sessions. A significant difference in asymmetry (*p* = 0.04) was found for impulse during the isometric squat when calculating asymmetry from the best trial. No other significant differences in asymmetry were indicated. Kappa coefficients revealed fair to substantial levels of agreement for asymmetry between test sessions, with the strongest consistency shown for the CMJ.

Data in Table 1 displays the within and between-session reliability for each test. A similar trend was observed during both test sessions, with the greatest variability seen during the isometric squat. Impulse in particular showed CV values > 10% on both limbs during both test sessions, potentially indicating that practitioners should be cautious of using this metric if using the unilateral isometric squat. Given the lower variability reported for this metric during bilateral isometric strength assessments [8,36], this represents a novel finding when considering a unilateral version of this test. In addition, results are comparable with previous literature using the unilateral isometric mid–thigh pull. Dos’Santos et al. [7] reported CV values of 10.5–11.6% for impulse in both professional rugby and collegiate athletes; thus, it would appear this metric may be subject to greater variability when assessed unilaterally. Furthermore, it is possible that greater familiarization is required in order to establish acceptable reliability for impulse during unilateral isometric strength assessments. Future research should aim to include additional testing sessions in an attempt to establish when variability has been reduced sufficiently (i.e., <10%). That said, relative reliability was good to excellent for all isometric squat metrics, with PF showing the strongest reliability throughout.

When considering the jump tests, within-session CV values were ≤8.1%, regardless of which jump test or metric analysed. Between-session variability showed a similar pattern, although jump height reported slightly greater variability (10.1–11.2%) during the unilateral drop jump in each leg. Relative reliability was good to excellent for all metrics during the unilateral CMJ, suggesting that jump height and PF are metrics with lower typical variability when quantifying unilateral vertical jump performance off a portable force platform. This serves as a useful finding for unilateral jump methods, given recent literature has validated the same portable force platform during bilateral jump testing [15]. The unilateral DJ showed good to excellent reliability for all metrics when quantified within-sessions; however, between-session reliability was reduced (moderate to good) and with slightly higher variability for jump height. In summary, the unilateral CMJ showed the strongest within and between-session reliability, with the unilateral DJ showing slightly larger variability for jump height. The DJ is a more technically challenging and less innate task when compared to the CMJ [13]; thus, it is likely that the slightly lower reliability scores can be attributed to the more advanced nature of the jump. Consequently, test familiarization is a key consideration for practitioners, especially when using more advanced test methods such as the DJ.

Data in Table 2 displays the mean test scores and inter-limb asymmetry values (calculated from the best trial and from averaging test scores on both the left and right sides). Differences were assessed between test sessions for the asymmetry scores using paired samples *t*-tests. The only significant difference between sessions was reported for impulse asymmetry during the isometric squat test (*p* = 0.04, effect size = −0.60), when calculated from the best trial. It is suggested that this is not necessarily a positive finding, given that our study used a test-retest design and no training intervention had been undertaken to warrant a change in asymmetry score. Furthermore, given that impulse also showed the greatest CV in all tests (Table 1), this further reiterates that practitioners may wish to be cautious of using this metric (when testing unilaterally) to quantify changes in inter-limb asymmetry following periods of training due its more variable nature. Consequently, practitioners could consider the average calculation method to be more favourable when reporting asymmetry data. This is supported in part by Lake et al. [26] who investigated whether the peak and mean force methods of calculating asymmetry agreed during a bilateral CMJ. Levels of agreement between methods were assessed using the Kappa coefficient and ranged from 0.67–0.72, representing ‘substantial’ levels of agreement. Whilst this may indicate a positive outcome, the authors proposed that given these values were not near perfect (i.e., Kappa values at or close to 1), that the two methods of quantifying asymmetry should not be used interchangeably. Furthermore, given the innate variability of asymmetry, an average of all trials may capture some of the inconsistency seen across trials (noting that if using unilateral test methods, the best score could be trial 1 on the left limb but trial 3 on the right limb).

Data in Table 4 displays the Kappa coefficients and accompanying descriptors for how consistently asymmetry favoured the same leg between test sessions, for each metric. The Kappa coefficient describes the proportion of agreement between two methods after any agreement by chance has been removed [33]. Levels of agreement were fair to substantial (0.29–0.64) for the isometric squat, substantial (0.64–0.66) for the CMJ and fair to moderate (0.36–0.56) for the DJ. Furthermore, it is interesting to note that greater levels of agreement appear to be associated with improved test reliability, noting that the CMJ showed the lowest CV values both within and between test sessions. These data indicate that the direction of asymmetry (i.e., how consistently the same leg scores higher between test sessions) varies considerably. Given this variable nature, it is suggested that individual data analysis is a key consideration for practitioners when monitoring inter-limb asymmetry (see Figure 2, Figure 3 and Figure 4). Despite recent literature highlighting poor levels of agreement for the same metric across tests [11], to the authors’ knowledge, this is the first study to report levels of agreement for the direction of asymmetry over more than a single test session. Thus, direct comparisons with previous research are not possible and requires further investigation using longitudinal study designs.

When interpreting the findings of the current study, practitioners should be aware of some wider considerations on the topic of asymmetry. Firstly, in addition to longitudinal monitoring, practitioners are advised to also consider more frequent monitoring in the short and long-term if asymmetry profiling is deemed appropriate for the athlete. Jump tests are commonly included during routine monitoring procedures [16,17] and practitioners may wish to consider asymmetry as a more regular line of investigation during such protocols. Doing so would enable practitioners to effectively determine trends in both the magnitude and direction of asymmetry. In turn, this may assist in the decision-making process when considering targeted training interventions for athletes. Secondly, testing modalities should always be considered within the context of athlete requirements when using data to help inform practice. The present study used unilateral tests to detect asymmetry; however, this may not always be appropriate. For example, in a sport such as weightlifting, virtually all movements are bilateral and if asymmetry analysis is deemed necessary, it is more than likely that test protocols should be conducted bilaterally to reflect the demands of the sport. Similarly, in team sports, many movement patterns occur unilaterally (e.g., sprinting, changing direction, kicking), in which case unilateral test protocols may be relevant. Practitioners are therefore advised to ensure that procedures are ecologically valid for the population in question, regardless of whether asymmetry is being investigated.

## 5. Conclusions

In summary, the magnitude of asymmetry appears to show significant differences between test sessions for the isometric squat when computing data from the best trial but not from an average of all trials. Given no training intervention was undertaken and no significant differences were found between test sessions when computing asymmetry from the average of all trials, it is suggested that the average method might be considered the most appropriate for calculating inter-limb differences. The direction of asymmetry appears highly variable; thus, individual data analysis is a strong consideration for practitioners and monitoring the direction of asymmetry may be more important than purely the magnitude when the purpose is to measure changes over time.

## Figures and Tables

**Figure 1 sports-07-00058-f001:**
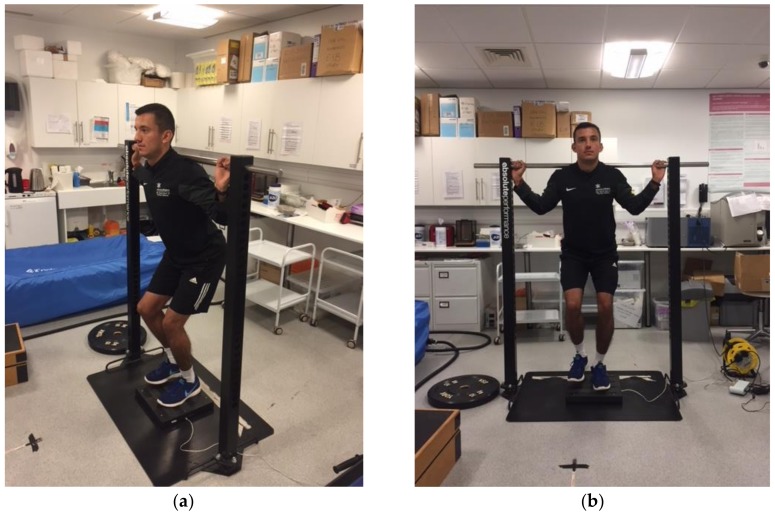
(**a**,**b**) Example positioning for the unilateral isometric squat protocol.

**Figure 2 sports-07-00058-f002:**
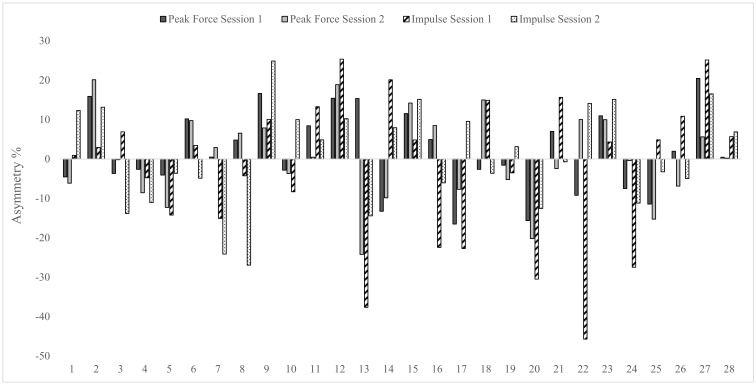
Individual asymmetry data for peak force and impulse during the unilateral isometric squat test in both test sessions. Above 0 indicates larger score on right leg and below 0 indicates larger score on left leg.

**Figure 3 sports-07-00058-f003:**
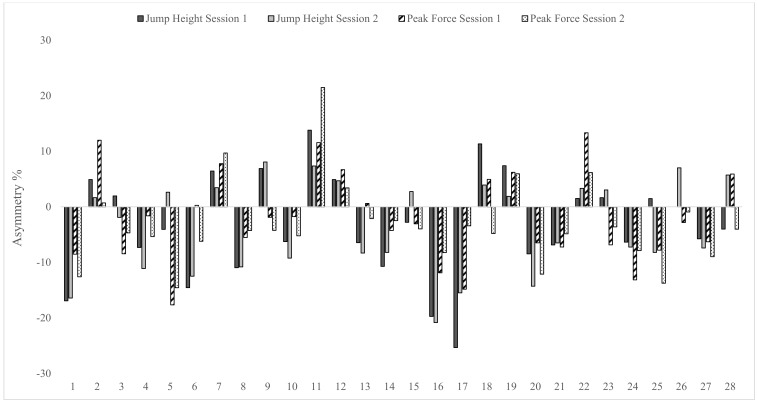
Individual asymmetry data for jump height and peak force during the unilateral countermovement jump test in both test sessions. Above 0 indicates larger score on right leg and below 0 indicates larger score on left leg.

**Figure 4 sports-07-00058-f004:**
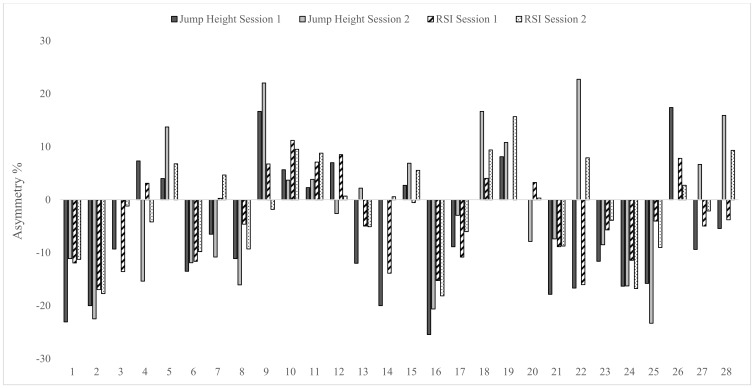
Individual asymmetry data for jump height and reactive strength index (RSI) during the unilateral drop jump test in both test sessions. Above 0 indicates larger score on right leg and below 0 indicates larger score on left leg.

**Table 1 sports-07-00058-t001:** Within and between-session test reliability data for the unilateral isometric squat, countermovement and drop jump tests.

Test/Metric		Test Session 1			Test Session 2			Between Session	
		ICC (95% CI)	CV (95% CI)	SEM	ICC (95% CI)	CV (95% CI)	SEM	ICC (95% CI)	CV (95% CI)
*Iso Squat*	PF-L (N)	0.94 (0.88–0.97)	5.4 (3.7–7.2)	107.5	0.96 (0.92–0.98)	4.9 (4.0–5.7)	78.8	0.93 (0.86–0.97)	6.4 (4.1–8.6)
	PF-R (N)	0.93 (0.87–0.96)	5.7 (4.2–7.2)	105.1	0.94 (0.89–0.97)	5.5 (4.1–6.9)	106.2	0.86 (0.72–0.93)	7.7 (5.6–9.8)
	Imp-L (N·s)	0.89 (0.80–0.94)	13.7 (10.0–17.4)	23.6	0.89 (0.81–0.95)	10.1 (8.0–12.3)	21.2	0.90 (0.80–0.95)	8.9 (6.2–11.5)
	Imp-R (N·s)	0.92 (0.85–0.96)	12.1 (9.5–14.8)	22.0	0.90 (0.83–0.95)	10.6 (8.1–13.1)	20.2	0.85 (0.70–0.93)	12.9 (9.4–16.4)
*SLCMJ*	JH-L (m)	0.89 (0.81–0.95)	4.8 (3.8–5.9)	0.01	0.87 (0.77–0.93)	4.2 (3.2–5.3)	0.01	0.82 (0.65–0.91)	3.7 (1.9–5.5)
	JH-R (m)	0.81 (0.68–0.90)	5.4 (4.1–6.7)	0.01	0.85 (0.74–0.92)	5.0 (4.0–6.0)	0.01	0.78 (0.59–0.89)	4.2 (2.5–5.8)
	PF-L (N)	0.89 (0.80–0.94)	5.8 (4.5–7.1)	67.7	0.93 (0.88–0.97)	4.9 (4.1–5.6)	42.9	0.85 (0.71–0.93)	6.2 (4.3–8.1)
	PF-R (N)	0.93 (0.87–0.96)	5.3 (4.2–6.4)	48.0	0.90 (0.82–0.95)	5.0 (3.9–6.2)	50.2	0.83 (0.67–0.92)	6.3 (4.4–8.2)
*SLDJ*	JH-L (m)	0.90 (0.82–0.95)	7.5 (5.4–9.7)	0.01	0.94 (0.89–0.97)	7.1 (4.3–9.9)	0.01	0.74 (0.49–0.87)	10.1 (6.3–13.9)
	JH-R (m)	0.90 (0.82–0.95)	8.1 (5.9–10.2)	0.01	0.93 (0.87–0.96)	6.8 (5.3–8.4)	0.01	0.75 (0.53–0.87)	11.2 (8.0–14.5)
	RSI-L	0.87 (0.77–0.93)	4.9 (3.7–6.0)	0.06	0.93 (0.88–0.97)	4.0 (2.6–5.3)	0.05	0.70 (0.37–0.86)	6.7 (4.5–8.9)
	RSI-R	0.91 (0.84–0.95)	4.7 (3.3–6.2)	0.06	0.88 (0.79–0.94)	5.9 (4.3–7.4)	0.07	0.84 (0.68–0.92)	5.1 (3.4–6.8)

ICC = intraclass correlation coefficient; CI = confidence intervals; CV = coefficient of variation; SEM = standard error of the measurement; Iso = isometric; PF = peak force; Imp = impulse at 0.3 s; N = Newtons; N·s = Newton seconds; L = left; R = right; SLCMJ = single leg countermovement jump; JH = jump height; m = meters; SLDJ = single leg drop jump; RSI = reactive strength index.

**Table 2 sports-07-00058-t002:** Mean test and asymmetry data (± SD) for test metrics reported from the best trial and average of all trials.

Test/Metric		Test Session 1				Test Session 2			
		Best Score	Asymmetry (%)	Average Score	Asymmetry (%)	Best Score	Asymmetry (%)	Average Score	Asymmetry (%)
*Iso Squat*	PF-L (N)	1597.0 ± 438.9	8.4 ± 6.8	1519.7 ± 414.8	8.6 ± 5.9	1631.3 ± 394.2	8.9 ± 6.9	1561.8 ± 392.3	9.0 ± 6.5
	PF-R (N)	1595.1 ± 397.3		1519.1 ± 382.4		1643.2 ± 433.4		1570.8 ± 424.6	
	Imp-L (N·s)	199.5 ± 71.2	15.5 ± 11.4	177.7 ± 69.3	14.5 ± 11.3	190.8 ± 64.0	9.6 ± 7.8*	174.5 ± 59.4	10.9 ± 6.7
	Imp-R (N·s)	192.9 ± 77.9		174.4 ± 75.0		191.9 ± 64.0		176.1 ± 61.6	
*SLCMJ*	JH-L (m)	0.21 ± 0.03	7.2 ± 6.1	0.20 ± 0.03	7.8 ± 5.9	0.22 ± 0.03	7.1 ± 5.0	0.21 ± 0.03	7.6 ± 4.9
	JH-R (m)	0.20 ± 0.03		0.19 ± 0.03		0.21 ± 0.03		0.20 ± 0.03	
	PF-L (N)	863.4 ± 204.0	7.5 ± 5.1	811.5 ± 177.6	7.1 ± 4.5	847.0 ± 162.3	6.6 ± 4.8	807.7 ± 156.5	6.6 ± 4.7
	PF-R (N)	830.8 ± 181.5		793.4 ± 174.0		818.6 ± 158.7		779.6 ± 141.8	
*SLDJ*	JH-L (m)	0.15 ± 0.03	10.1 ± 8.7	0.14 ± 0.03	11.1 ± 6.9	0.14 ± 0.04	10.7 ± 8.6	0.13 ± 0.04	10.8 ± 7.5
	JH-R (m)	0.14 ± 0.03		0.13 ± 0.03		0.13 ± 0.04		0.13 ± 0.04	
	RSI-L	1.31 ± 0.17	8.1 ± 4.8	1.25 ± 0.18	7.5 ± 5.1	1.23 ± 0.20	7.3 ± 4.7	1.19 ± 0.20	7.4 ± 5.2
	RSI-R	1.26 ± 0.20		1.21 ± 0.20		1.23 ± 0.20		1.17 ± 0.20	

* Significantly different from asymmetry score (calculated from the best trial) in test session 1 (*p* = 0.04). Iso = isometric; PF = peak force; Imp = impulse at 0.3 s; N = Newtons; N·s = Newton seconds; L = left; R = right; SLCMJ = single leg countermovement jump; JH = jump height; m = meters; SLDJ = single leg drop jump; RSI = reactive strength index.

**Table 3 sports-07-00058-t003:** Between-session effect size data (95% confidence intervals) for test and asymmetry scores using both methods of calculation.

Test/Metric		Best Score	Asymmetry % (from Best Score)	Average Score	Asymmetry % (from Average Score)
*Iso Squat*	PF-L (N)	0.08 (0.80 to −0.63)	0.08 (0.79 to −0.64)	0.10 (0.82 to −0.61)	0.07 (0.79 to −0.64)
	PF-R (N)	0.12 (0.83 to −0.60)		0.13 (0.84 to −0.59)	
	Imp-L (N·s)	−0.13 (0.59 to −0.85)	−0.60 (0.14 to −1.33)	−0.05 (0.67 to −0.77)	−0.38 (0.34 to −1.10)
	Imp-R (N·s)	−0.01 (0.70 to −0.73)		0.03 (0.74 to −0.69)	
*SLCMJ*	JH-L (m)	0.33 (1.05 to −0.39)	−0.03 (0.69 to −0.74)	0.33 (1.05 to −0.39)	−0.03 (0.68 to −0.75)
	JH-R (m)	0.33 (1.05 to −0.39)		0.33 (1.05 to −0.39)	
	PF-L (N)	−0.09 (0.63 to −0.81)	−0.18 (0.53 to −0.90)	−0.02 (0.69 to −0.74)	−0.11 (0.61 to −0.82)
	PF-R (N)	−0.07 (0.64 to −0.79)		−0.09 (0.63 to −0.80)	
*SLDJ*	JH-L (m)	−0.28 (0.44 to −1.00)	0.07 (0.79 to −0.64)	−0.28 (0.44 to −1.00)	−0.04 (0.67 to −0.76)
	JH-R (m)	−0.28 (0.44 to −1.00)		0.00 (0.72 to −0.72)	
	RSI-L	−0.43 (0.29 to −1.15)	−0.18 (0.54 to −0.90)	−0.32 (0.40 to −1.04)	−0.03 (0.69 to −0.74)
	RSI-R	−0.15 (0.57 to −0.87)		−0.20 (0.52 to −0.92)	

Iso = isometric; PF = peak force; Imp = impulse at 0.3 s; N = Newtons; N·s = Newton seconds; L = left; R = right; SLCMJ = single leg countermovement jump; JH = jump height; m = meters; SLDJ = single leg drop jump; RSI = reactive strength index.

**Table 4 sports-07-00058-t004:** Kappa coefficients and descriptive levels of agreement showing how consistently asymmetry favours the same leg between test sessions for the unilateral isometric squat, countermovement and drop jump tests.

Test/Metric		Kappa Coefficient	Descriptor
*Isometric Squat*	Peak Force	0.64	Substantial
	Impulse at 0.3s	0.29	Fair
*SLCMJ*	Jump Height	0.64	Substantial
	Peak Force	0.66	Substantial
*SLDJ*	Jump Height	0.36	Fair
	Reactive Strength Index	0.56	Moderate

SLCMJ = single leg countermovement jump; SLDJ = single leg drop jump.

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
