# Peer review of "Using Unilateral Strength, Power and Reactive Strength Tests to Detect the Magnitude and Direction of Asymmetry: A Test-Retest Design"

_sports, 2019, doi:10.3390/sports7030058_

Round 1
Reviewer 1 Report
This manuscript may contribute to the existing unilateral literature that current exists and the authors should be commended for their work. There are several specific concerns that I feel should be addressed.
-Lines 59-60: A recent paper by Lake et al. (Do the peak and mean force methods of assessing vertical jump force asymmetry agree? Sports Biomechanics, ahead of print, 2018) should be included here. The consistency of jump asymmetry is certainly a concern when it comes to assessing an individual's asymmetry properties.
-Line 118: Why was the 0.3 s time interval chosen? A rationale for its inclusion is needed.
-Line 129: Further clarification is needed here. Are the weight plates touching the force plates? This may throw some numbers off.
-Line 140: How was the propulsive phase of the jump identified? Further detail is needed here.
-Line 147: What was the rationale of using the flight time method to calculate jump height rather than take-off velocity?
-Figure 2: How can some of the conclusions of the current study be drawn due to the consistently inconsistent asymmetry within individuals? By my count, 16 of the 28 participants exhibited asymmetry in different directions. Further discussion is needed. The same can be said about Figures 3 and 4 as well.
-Line 225: Were there any instructions regarding the foot placement of individuals on the force plates? This may modify the consistency of asymmetry.
-Line 268: Again, Lake et al., 2018 may be discussed here.
Author Response
Thank you for taking the time to review our manuscript.
We have responded to your comments and detailed revisions on the attached document.
Kind regards

Reviewer 2 Report
General Comments to the Authors: Overall the authors have presented an interesting paper which examines a unique test. There are couple of things that need to be considered and revised, which I have noted in the specific comments.
Specific Comments to the Authors
Line Comment
15 reword: t-tests were used to
19 I would report the effect size here
38 reword: force-time curve
89 reword: within the preceding three months.
89-90 Make sure you are clear that the participants read and signed all informed consent documents. This is an important statement that is clear.
96-97 Do you think randomizing could have biased some of your data? What I mean was the same order of testing used on each testing session per a given subject? As you are looking at reliability and lets say on day 1 you did Iso Squat then jump and on Day 2 you did Jump then Iso Squat we could argue that the Iso Squat potentiated the jump and this could result in changes in your result that are not related to the test, but to the order of the test? Clarification on this is warranted.
98 What was the proximity of the familiarization session to the testing session?
107 ‘Iso Rig’ seems a bit jargon like. Is it a custom Isometric testing system? I would be less jargon like in your wording.
112-115 Why hoover the leg? Why not use a Bulgarian or Single leg squat stance?
149 I would suggest reporting effect size statistics, specifically Hedge’s g
154-157 I would report the 95% CI for both the CV and ICC
172-173 with an upper limit of 13.7% for the CV I might argue that based upon your criteria on line 155 (i.e.<10%) that your test is not acceptable? It would be useful to see the 95% CI for both the CV and ICC to see how they relate to reliability cut-offs
206-207 Figure 2: I would suggest reporting these data with stacked scatter plots so we can see the spread of the subjects
209-211 Figure 3: I would suggest reporting these data with stacked scatter plots so we can see the spread of the subjects
212-214 Figure 4: I would suggest reporting these data with stacked scatter plots so we can see the spread of the subjects
226 Reword: Data in table 1 displays the
227 Could the greater variability be a result of the hoover position used?
235-237 I might argue that the lack of reliability is related to the unilateral squat position used. If you look at the paper by Urquhart et al. 2015 the split squat 1RM was found to be very reliable (Cv= 2.5%; ICC = 0.97). By extension it is likely that the isometric split squat would also be highly reliable – this should be considered in your write up
Urquhart BG, Moir GL, Graham SM, and Connaboy C. Reliability of 1RM Split-Squat Performance and the Efficacy of Assessing Both Bilateral Squat and Split-Squat 1RM in a Single Session for Non-Resistance-Trained Recreationally Active Men. J Strength Cond Res 29: 1991-1998, 2015.
255 Data is reported in Table 2 – please revise accordingly.
Author Response
Thank you for taking the time to review our manuscript.
General Comments to the Authors:
Overall the authors have presented an interesting paper which examines a unique test. There are couple of things that need to be considered and revised, which I have noted in the specific comments.
Thank you to the reviewer for their time and effort in critiquing our manuscript, your efforts are very much appreciated and we have provided a point-by-point response below to your suggestions.
Specific Comments to the Authors
Line Comment
15: reword: t-tests were used to
Amended as per your suggestion
19: I would report the effect size here
Effect size added here as per your request
38: reword: force-time curve
Amended in line with your suggestion
89: reword: within the preceding three months.
This has actually been amended to “in the preceding three months” as per the suggestion from one of the other reviewers.
89-90: Make sure you are clear that the participants read and signed all informed consent documents. This is an important statement that is clear.
Fair point, some small wording amendments have been made to provide a clearer message here.
96-97: Do you think randomizing could have biased some of your data? What I mean was the same order of testing used on each testing session per a given subject? As you are looking at reliability and let’s say on day 1 you did Iso Squat then jump and on Day 2 you did Jump then Iso Squat we could argue that the Iso Squat potentiated the jump and this could result in changes in your result that are not related to the test, but to the order of the test? Clarification on this is warranted.
This is a good point and thank you for raising it. Yes, although test order was randomized for each participant, they did actually follow the same test order for both test sessions, so there was no risk of changes in results occurring for that reason as per your query. Additional detail of this has been provided as per your request, good suggestion.
98: What was the proximity of the familiarization session to the testing session?
72 hours – this has been added in as per your request.
107: ‘Iso Rig’ seems a bit jargon like. Is it a custom Isometric testing system? I would be less jargon like in your wording.
Fair point, we have changed this term to “isometric testing system” as per your suggestion, thank you for this.
112-115: Why hoover the leg? Why not use a Bulgarian or Single leg squat stance?
The authors did provide a rationale for the non-stance limb position – it was an attempt to ensure that the hips remained level; thus, aiding in balance and stability. In our opinion, the use of a Bulgarian stance would not be truly unilateral and is therefore not entirely comparable given the foot contact of the rear limb.
149: I would suggest reporting effect size statistics, specifically Hedge’s g
As per your request, we have added in between-session Hedge’s g effect size data for both test and asymmetry scores, calculated from both the best and average of all trials methods. This is now shown in Table 3 and we believe that this needs to be presented as a separate table because a) there are four columns of effect size data to present; thus, integrating it into an existing table becomes too crowded and b) the effect sizes are only for between-session, which makes it almost impossible to integrate into one of the other tables. Thank you for the suggestion here, we believe this is a good addition.
154-157: I would report the 95% CI for both the CV and ICC
95% CI’s have been added in for the CV as per your suggestion, thank you for this.
172-173: with an upper limit of 13.7% for the CV I might argue that based upon your criteria on line 155 (i.e.<10%) that your test is not acceptable? It would be useful to see the 95% CI for both the CV and ICC to see how they relate to reliability cut-offs
The 95% CI’s have been calculated as per your request and as the data shows, the lower boundaries are < 10%; thus, we feel that the chosen language used in the manuscript pertaining to practitioners being “cautious” is appropriate here.
206-207: Figure 2: I would suggest reporting these data with stacked scatter plots so we can see the spread of the subjects
209-211: Figure 3: I would suggest reporting these data with stacked scatter plots so we can see the spread of the subjects
212-214: Figure 4: I would suggest reporting these data with stacked scatter plots so we can see the spread of the subjects
Thank you for the suggestion here re: the figures. Respectfully, the authors believe the bar charts are a better representation of the data in this scenario because a scatter plot might potentially make it harder to compare participant scores for the same metrics as we are also depicting the direction of asymmetry, not just the magnitude. Equally, one of the other reviewers commented on the appropriateness of the figures and suggested we amended the shading of the bars to further aid with clarity for the reader, which we have done. Finally, the presentation of this data is in line with recent publications on the topic of asymmetry; see example references below if interested. Thank you for considering our stance here.
Bishop et al. (2018). Interlimb asymmetries: The need for an individual approach to data analysis. J Strength Cond Res.
Maloney et al. (2017). Do stiffness and asymmetries predict change of direction speed performance? J Sports Sci.
Dos’Santos et al. (2017). Asymmetries in single and triple hop are not detrimental to change of direction speed. J Trainology.
Bishop et al. (2018). Vertical and horizontal asymmetries are related to slower sprinting and jump performance in in elite youth female soccer players. J Strength Cond Res.
Bishop et al. (2019). Drop jump asymmetry is associated with reduced sprint and change-of-direction speed performance in adult female soccer players. Sports.
226: Reword: Data in table 1 displays the
Amended as per your suggestion
227: Could the greater variability be a result of the hoover position used?
Thank you for the comment. This could be a possibility; however, it’s worth noting that relative reliability (ICC) was pretty strong and that peak force was also fine as a metric both within and between-sessions. The authors believe that it comes down to impulse being a less reliable and noisier metric than peak force which a) we have alluded to in the discussion section and, b) is in agreement with similar research using the unilateral IMTP, which we have also acknowledged.
235-237: I might argue that the lack of reliability is related to the unilateral squat position used. If you look at the paper by Urquhart et al. 2015 the split squat 1RM was found to be very reliable (CV= 2.5%; ICC = 0.97). By extension it is likely that the isometric split squat would also be highly reliable – this should be considered in your write up
Urquhart BG, Moir GL, Graham SM, and Connaboy C. Reliability of 1RM Split-Squat Performance and the Efficacy of Assessing Both Bilateral Squat and Split-Squat 1RM in a Single Session for Non-Resistance-Trained Recreationally Active Men. J Strength Cond Res 29: 1991-1998, 2015.
Thank you for taking the time to consider some alternative options for us within the context of our discussion, this is very much appreciated. Respectfully however, the authors do not feel this is directly comparable given that the Urquhart study investigated 1RM values, not peak force or impulse. In no way are we demeaning the value of the Urquhart study; however, given the variables/metrics are different, we do not see how this is a fair comparison. Equally, the split squat used in that study is not truly unilateral given both feet are on the floor, which again we would argue is not comparable to the methods used in our study. In addition, our discussion has also had a small wording amendment (as per the request from one of the other reviewers), which provides a slightly stronger statement re: using impulse during this test. In short, wording has been amended from “should be mindful” to “should be cautious” which should provide the reader with enough sway to convince them that this may not be an appropriate metric to monitor during the unilateral isometric squat. In contrast though, peak force appears to be fine and we have also stated that as well. Thank you for the feedback and suggestion here, appreciated.
255: Data is reported in Table 2 – please revise accordingly.
Amended as per your suggestion and the same has been done for the following paragraph when discussing Table 4.
Reviewer 3 Report
Great job to the authors! I look forward to reading your revisions and hearing your comments. My comments are listed line-by-line below:
Line 15: add “were” after “t-tests”
Line 30: add “differences in” after “refers to”
Line 32: the word “trait” implies something unchangeable or mostly genetic.
Line 33: it seems like citations 3-5 are supporting the first clause of this sentence but not the second. If this is so, move the citation forward to make this clear.
Line 31-36: specify that you are speaking about strength and jump tests of asymmetry. This is not clear unless the reader shifts back-and-forth between the text and the references.
Line 36: A more likely reason is because of the high degree of similarity that CMJ and DJ have with sport-specific movements. Ease of implementation is important but secondary.
Line 37: “provided that” or “provided”
Lines 37-39: This sentence seems paradoxical. You are essentially saying “as long as coaches have force plates, then analysis via force plates is a viable option.” Consider rephrasing or rewriting
Line 41: When you say “force and impulse”, do you mean peak force, mean force, rate of force, force at a certain time point? Same question for impulse? Be more specific here. Gathercole et al. (whom you cited) list typical and alternative CMJ variables of interest.
Line 43: “with” respect to
Line 48: “directly compared” instead of “investigated a direct comparison of”
Line 56: This sentence says the same thing as the previous sentence. Consider removing
Line 56: The phrase “Further to this,” is not grammatically correct. The sentence is fine without a transitioning word or phrase at the beginning
Line 57-58: I’m not sure you need to define “direction of asymmetry”. It is self-evident that an asymmetry has a direction.
Line 59: the variability of the direction of asymmetry, on the other hand, is not self-evident. Perhaps this is a good second sentence for this paragraph.
Line 79: change “established” to “quantified”. The way it is used here “established” implies that you found bias, but of course a report of that finding would be reserved for the results section.
Line 80: The part of the sentence starting with “noting that data…” should be put in the intro. The methods section should be for your own methods with supporting citations. It’s an important point but the methods section is strictly a recipe to reproduce the present study.
Line 85: perhaps “recreational soccer and rugby athletes” ?
Line 89: “or in the preceding”
Line 128: “and were require to remain”
Line 129: “on the ground on each side”
Lines 130-131: “This was maintained for all jump tests.” This sentence is likely not necessary given that you just devoted a sentence to detailing the purpose of the weights as part of the testing setup.
Line 143: “in-line” or “based on”
This is more out of curiosity, but is it not possible to calculate drop jump height based on impulse-momentum method as well? Or did you use flight time to be consistent with your RSI calculation?
Line 155: I could be wrong, but I think it might be a better formatting choice to put citation 16 outside of the parentheses
General comment: Excellent decision to include the individual asymmetry data.
General comment: The individual data may be easier to interpret if the shading of the force data was similar, and the shading of the impulse data was similar. Due to the density of the columns it is difficult to visually assess without going column-by-column. Maybe force data could be shades of gray, and impulse data could be stripes and dots? Just a suggestion for ease-of-reading
Line 229: “may wish to be mindful”. As a practitioner I don’t know if you are saying that I should look closely at this metric or that I should be wary of using it. Perhaps be more specific as to what we should be mindful of.
Line 233: change “using” to “in”. As sport scientists our participants are also our primary beneficiaries.
Line 242: change “on both legs” to “in each leg”.
General comment: In your discussion, a lot of space is used to state or restate your results. My suggestion is to take any sentence stating a result and move it to the results section unless it is necessary to remind the reader what you’re referencing.
General comment: As a practitioner using CMJ and IMTP testing frequently, I am curious as to why I would use these unilateral tests rather than performing all tests bilaterally but on a dual platform? Unilateral testing requires twice as much time from the athlete, storage in your monitoring system, and time from sport scientists to analyze. I think that your findings regarding the between- and within-session variability of limb asymmetry is important and timely, but are unilateral tests really the best way for the practitioner to assess this? I know you’re not necessarily making a case for this, but perhaps mention that the same findings may hold true for bilateral tests of asymmetry as well (as long as the literature corroborates this).
General comment: your conclusions follow from the results and align with the literature, but seem to be conclusions that we were already aware of. Averaged data is more consistent than individual trials, we should pay attention to which direction asymmetry occurs, and in an athlete monitoring program certain variables (such as limb asymmetry) are monitored at the level of the individual. What else does this particular analysis have to offer?
Overall: Really an excellent study design and an important contribution to the athlete monitoring and testing literature. I would like to see what more insight you can dig up from your findings, and how it might impact the practitioner. Given the variability of limb asymmetry direction that you demonstrated, how might our testing methods and test analysis change? Would you recommend establishing a limb asymmetry baseline from which to compare future tests? Perhaps trend analysis? Based on your data was there a threshold within which the direction of asymmetry was more likely to change? Are the bilateral tests that most practitioners employ useful for detecting these asymmetries?
Author Response

(The authors gave the same response as above.)

Round 2
Reviewer 1 Report
The authors have addressed all of my primary concerns.
Reviewer 3 Report
Excellent responses. I enjoyed reading your manuscript a second time with these changes. You have done a good job of either incorporating all of the proposed changes, or stating a clear and concise rebuttal that I am satisfied with. I believe that this is a timely contribution to the literature as a colleague and I were recently discussing the topic of "asymmetry thresholds" and injury in regards to our university's athlete monitoring program. Thank you for contributing to that discussion.